# A Method for Quantifying Uncertainty in Spatially Interpolated Meteorological Data with Application to Daily Maximum Air Temperature

Conor T. Doherty[1], Weile Wang[1], Hirofumi Hashimoto[1,2], Ian G. Brosnan[1]

[1]NASA Ames Research Center, Moffett Field, CA, 94035, USA
[2]California State University Monterey Bay, Seaside, CA, 93955, USA

*Correspondence to*: Conor T. Doherty (conor.t.doherty@nasa.gov)

**Abstract.** Uncertainty is inherent in gridded meteorological data, but this fact is often overlooked when data products do not provide a quantitative description of prediction uncertainty. This paper describes, applies, and evaluates a method for quantifying prediction uncertainty in spatially interpolated estimates of meteorological variables. The approach presented here, which we will refer to as DNK for "detrend, normal score, krige," uses established methods from geostatistics to produce not only point estimates (i.e., a single number) but predictive distributions for each location. Predictive distributions quantitatively describe uncertainty in a manner suitable for propagation into physical models that take meteorological variables as inputs. We apply the method to interpolate daily maximum near-surface air temperature (Tmax), then validate the uncertainty quantification by comparing theoretical versus actual coverage of prediction intervals computed at locations where measurement data were held out from the estimation procedure. We find that, for most days, the predictive distributions accurately quantify uncertainty and that theoretical versus actual coverage levels of prediction intervals closely match one another. Even for days with the worst agreement, the predictive distributions meaningfully convey the relative certainty of predictions for different locations in space. After validating the methodology, we demonstrate how the magnitude of prediction uncertainty varies significantly in both space and time. Finally, we examine spatial correlation in predictions and errors by using conditional Gaussian simulation to sample from the joint spatial predictive distribution. In summary, this work demonstrates the efficacy and value of describing uncertainty in gridded meteorological data products using predictive distributions.

## 1 Introduction

Interpolated meteorological data products are widely used in the geosciences, but relatively little attention is paid to the errors they contain. For example, when studying terrestrial fluxes of carbon, water, and energy over a large spatial domain (e.g., $\geq 100$ km$^2$), it is necessary to work with gridded meteorological data. Ground-based weather stations may be sparse or only cover a small fraction of the study area so gridded estimates, rather than station measurements, of meteorological variables are used by models of land surface processes (Zeng et al., 2020; Volk et al., 2024). In many gridded data products, the values are point estimates (i.e., a single number rather than a range or distribution). When given only point estimates, data users do not know, and cannot propagate, the uncertainty in the meteorological inputs to their model. While users may refer to point estimate

accuracy statistics for the data product, these statistics only capture errors at locations where measurements are available. For applications that are particularly sensitive to meteorological inputs, such as evapotranspiration modeling, uncertainty in gridded data can contribute significantly to downstream model errors (Doherty et al., 2022). While geostatistical uncertainty quantification is standard practice in other domains like mining (Rossi and Deutsch, 2014), oil and gas exploration (Pyrcz and Deutsch, 2014), and hydrogeology (Kitanidis, 1997), these methods are not as widely used in popular near-surface meteorological data products. Understanding uncertainty in gridded meteorological data is necessary to evaluate the robustness of scientific findings, especially when designing and implementing public policy based on those findings (Morgan and Henrion, 1990).

One approach to producing gridded near-surface meteorological data is statistical interpolation, where gridded values are estimated by interpolating between measurements at ground-based weather stations. For North America, DayMET (Thornton et al., 1997; Thornton et al., 2021) and PRISM (Daly et al., 2008), which produce estimates of several meteorological variables on fine spatial grids (~1 km$^2$), are widely used statistical interpolation products. A related product, NEX-GDM (Hashimoto et al., 2019), uses machine learning and a wide range of inputs to produce high resolution gridded meteorological values. For Europe, E-OBS (Cornes et al., 2018) is an interpolated data product with 12 km spatial resolution. Regarding uncertainty, Daly et al. (2008) describes a method for creating prediction intervals, but the resulting maps are not publicly distributed. Thornton et al. (2021) includes an extensive accuracy assessment using cross validation, but the methodology does not produce spatially resolved uncertainty estimates. The E-OBS methodology is the most similar to this work in terms of modeling the data as a Gaussian random field and producing predictive distributions for each grid cell. However, there are important differences in the data processing, the effects of which may explain some of the results in Cornes et al. (2018). We address this topic in the Discussion. Spatial machine learning methods including Quantile Random Forest (QRF) can also be used to produce prediction intervals for uncertainty quantification (Hengl et al., 2018; Milà et al., 2023). However, this approach does not take into account spatial correlation and is sometimes combined with other geostatistical methods that do (Milà et al., 2023). Finally, Bayesian methods using Gaussian Markov Random Fields (GMRF) have also been used to perform probabilistic interpolation of meteorological data. For example, Fioravanti et al. (2023) applied these methods to air temperature (but did not quantitatively validate the uncertainty quantification) and Ingebrigtsen et al. (2014) applied them to precipitation data. In future work, it would be instructive to compare predictive distributions produced using QRF and Bayesian GMRF-based methods with those produced using the more classical geostatistical methods described in this work.

Another class of approaches to producing gridded data products is data assimilation, which combine dynamic physical models with data-driven adjustments. This work is focused on statistical interpolation rather than data assimilation, but we give a brief overview of the latter for context. A wide range of data assimilation products are available including regional products like RTMA (De Pondeca et al., 2011), CONUS404 (Rasmussen et al. 2023), and global ones like MERRA2 (Gelaro et al., 2017) and ERA5 (Hersbach et al., 2020; Bell et al., 2021). Some assimilation products, like ERA5, express uncertainty using an ensemble of model runs, where a greater magnitude of spread in the ensemble is taken to indicate greater uncertainty. However, the computational expense of large-scale climate simulations generally means that the resulting data products have

relatively coarse spatial resolution (31 km horizontal resolution for ERA5) and ensembles that are not large enough (tens of ensemble members) to characterize stable empirical distributions. In contrast, the approach described in this work is computationally efficient enough to be run at fine spatial resolution over large areas while also giving a robust description of the predictive distribution.

In this paper we present and analyze a statistical method to produce spatially and temporally resolved uncertainty quantification and apply it to the interpolation of daily maximum near-surface air temperature (Tmax). We will refer to the approach for estimation and uncertainty quantification as DNK for "detrend, normal score, krige." The basic approach of DNK is well-established in geostatistics, appearing in textbooks such as Olea (1999), Goovaerts (1997), and others. While kriging and related spatial regression methods have previously been used for meteorological data interpolation, they are most

commonly used to produce gridded point estimates. A central component on this work is to test the validity of predictive distributions generated using DNK and, as such, their utility for uncertainty quantification. Uncertainty is not intrinsic to macro-scale physical phenomenon but rather is a property of the combination of data and a model (Goovaerts, 1997), which means that there is not an objective "correct" predictive distribution for a given unknown value. However, we can assess the validity of a collection of predictive distributions, in aggregate, by testing the rate at which true measurement data fall within

prediction intervals relative to those intervals' theoretical coverage. If the validity of predictive distributions can be established, then DNK can accurately quantify uncertainty in gridded meteorological data.

## 2 Methods

### 2.1 Input Data

This study uses two sets of input data: daily maximum air temperature at 2 m (Tmax) and elevation. Tmax data are provided

by Thornton et al. (2022) for stations in the Global Historical Climatology Network (GHCN) (Menne et al., 2012), a database of measurement data from ground-based weather stations across the world. The GHCNd (daily) data are processed as described in Thornton et al. (2021) to correct for temperature sensor biases and inconsistencies in time of observation. Figure 1A shows the spatial distribution of weather stations across the study area. We use data from 2022 to conduct the validation study. In 2022, the number of weather stations within the California state boundary ranges between 524 and 542 stations depending on

the day of year. Data from stations within the state boundaries are used as ground truth for validation. Data from stations outside the study area contribute to predictions at locations near the boundary but these stations are not, themselves, used as validation locations. Elevation data are sourced from a digital elevation model (DEM) with 90 m resolution for the western United States (Hanser, 2008). The DEM is clipped to the boundaries of the study area and then resampled using mean resampling to a grid with 1 km$^2$ grid cells. Figure 1B shows example Tmax time series for three stations from different types

of regions: "USC00045795" (blue, coastal), "USW00053119" (orange, inland), and "USS0019L45S" (green, mountain).

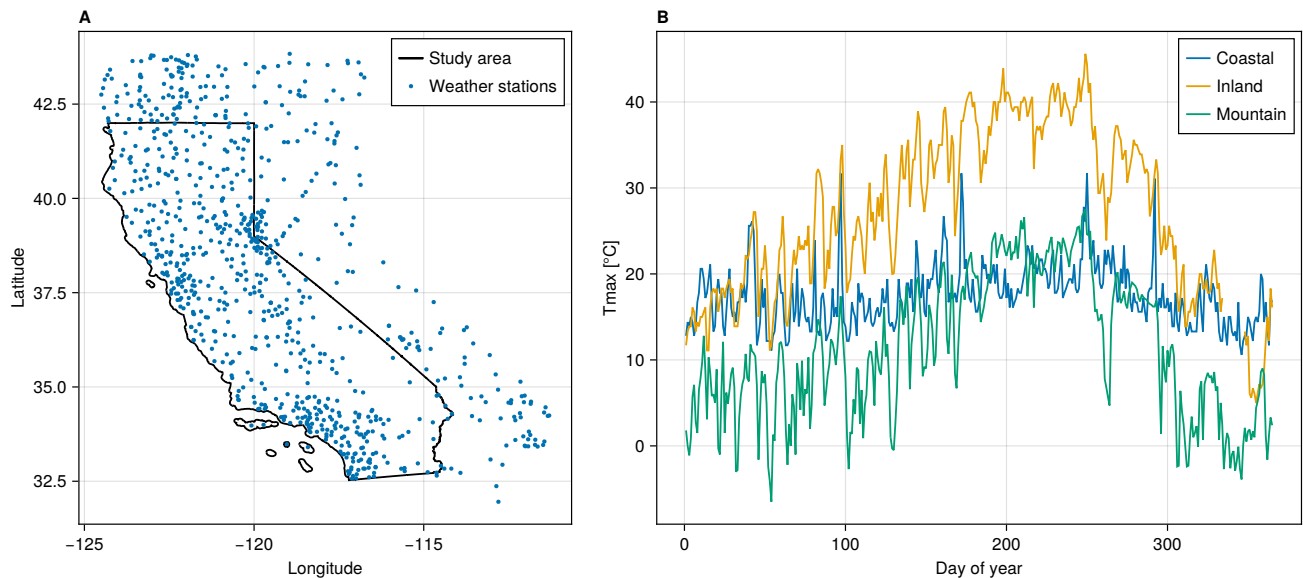

**Figure 1:** Study area, locations of weather stations, and sample Tmax time series. Panel A shows the study area and weather station locations. Black lines mark the bounds of the study area (the state of California). Blue dots mark the locations of GHCN weather stations that were active in 2022. Panel B shows sample time series for three weather stations, each from a different type of location.

## 2.2 Probabilistic Interpolation Procedure

The interpolation procedure, which we summarize as "detrend, normal score, krige" (DNK), combines established methods from geostatistics to produce predictive distributions. We briefly summarize the procedure here, before providing greater detail in the following sub-sections. At a high level, the data are modeled as being observations of a stationary Gaussian random field. The "detrend" and "normal score" steps are transformations applied to make the data better satisfy the assumptions of this mathematical model. The procedure consists of the following steps:

1.  Detrend: estimate and subtract spatial trends from measurement data
2.  Normal score: apply quantile-to-quantile mapping transforming observed empirical distribution to standard normal distribution
3.  Krige: produce marginal (or joint) predictive distributions at prediction locations using Ordinary Kriging (or Conditional Gaussian Simulation)
4.  Inverse normal score transform: apply the inverse of the mapping from step (2) to the output of step (3), transforming predicted values back to the "original" (detrended) distribution
5.  Add back the trend: estimate spatial trends at prediction locations and add them to the output of step (4) to produce final predictive distributions

The procedure is also represented graphically in Fig. 2, where each step in the previous list corresponds to a transition from one column to the next (moving from left to right). We will refer to Fig. 2 in the following sub-sections. All calculations for this study were performed using the Julia programming language (Bezanson et al., 2017), and we make significant use of the GeoStats.jl package (Hoffimann, 2018) in particular. All package names and versions that were used can be found in the Manifest.toml file provided (see Code Availability).

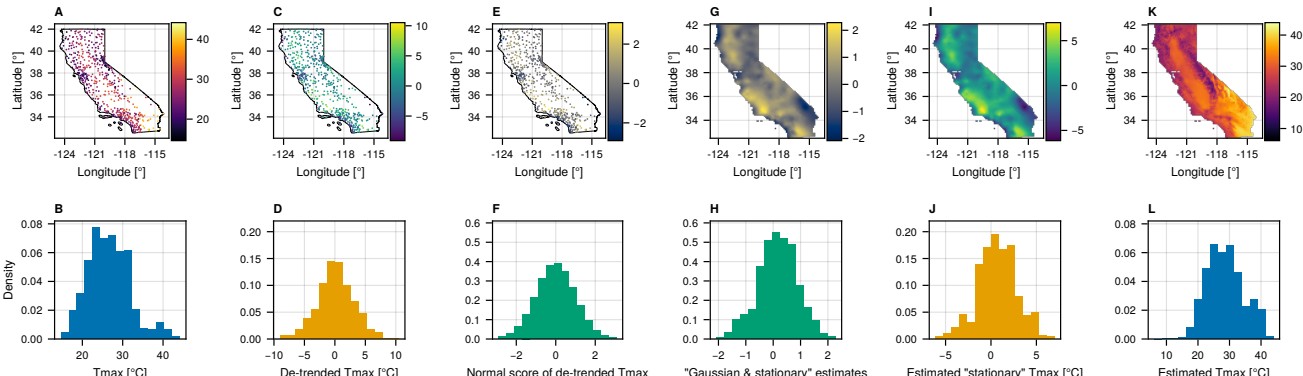


**Figure 2:** Illustration of data processing steps. Each column corresponds to a step in the processing and estimation pipeline. The top row shows point (panels A, C, E) and gridded (panels G, I, K) data values. The bottom row shows histograms of the data at each processing step. Panels A-B show the measured Tmax values. Panels C-D show the Tmax values after spatial trends have been estimated and subtracted. Panels E-F show the detrended Tmax data after the normal score transform has been applied. Panels G-H show gridded estimates produced

by OK in the detrended and Gaussian space. Panels I-J show gridded estimates after reverting the normal score transform using quantile information from the distributions in panels C-D. Panels K-L show the final gridded estimates after re-adding spatial trends.

### 2.2.1 Detrending

The first transformation models and then subtracts local spatial trends. The purpose of trend modeling is to identify and remove variation at coarse spatial scales that would otherwise make the data nonstationary. We use the term "trend" to refer both to

large scale variation in longitude-latitude space and variation due to change in elevation (lapse rate). While some prior approaches assume a fixed lapse rate (Hart et al., 2008), we allow lapse rate to vary in space (Thornton et al., 1997; Thornton et al. 2021; Daly et al., 2008; Cornes et al. 2018). Local trends are estimated by regressing Tmax values on spatial coordinates and elevation:

$$Tmax_i = x_i + y_i + z_i + \epsilon_i$$


$$( 1 )$$

where $x_i$ and $y_i$ are the spatial coordinates and $z_i$ is the elevation at weather station $i$. The residual $\epsilon_i$, calculated as the observation minus the fitted value from the trend model, is taken as the detrended Tmax value. Detrending is performed locally using a 100 km search radius around each weather station. Detrended station-level Tmax values are shown in Fig. 2B-C. A similar procedure is applied to "add back" the trend after estimation: the trend parameters are estimated at locations centered

on each grid cell, again using data from weather stations within 100 km. The cell-wise trend value is calculated using the regression parameter estimates and the corresponding coordinates of the cell and elevation from a digital elevation model. The gridded estimates with the trend added back is shown in Fig. 2K-L. The purpose of detrending is not to explain all variation in Tmax values. Rather, the purpose is to control for large scale variation such that the residuals, after detrending, can be modeled as a random field whose mean does not vary deterministically in space. While it could be useful to incorporate other covariates

(e.g., distance to the ocean), doing so would likely require a nonlinear trend model.

### 2.2.2 Normal Score Transformation

The second transformation, a "normal score transformation," transforms the data to be approximately Gaussian. The transformation is done by mapping quantiles of the empirical distribution of detrended Tmax values to the corresponding quantiles of a standard normal distribution. The transformed station-level data are shown in Fig. 2C-D. The quantile

information from the original empirical distribution is saved so that the inverse transformation can be applied after estimation. The gridded estimates, after reverting the normal score transform, are shown in Fig. 2I-J. This transformation is implemented in TableTransforms.jl. Inclusion of this step is one place where our approach differs from the E-OBS methodology (Cornes et al., 2018) and could explain some of the differences in results. We address this difference in the Discussion.

### 2.2.3 Estimation of Marginal Predictive Distributions using Ordinary Kriging

The primary estimation method we consider is Ordinary Kriging (OK) (see e.g., Goovaerts, 1997; Olea, 1999; Anderes, 2012), which gives analytical solutions for both a point estimate and the variance of a predictive distribution. For a random field that is covariance stationary and Gaussian, the OK prediction mean and variance completely characterize the predictive distribution. In general, spatially distributed Tmax data satisfy neither assumption, which is why we first apply detrending and normal score transformations. We apply OK locally at each prediction location using measurement data within a 100 km radius

of the location in the estimation. Figure 2G-H shows an example of the gridded estimates produced by OK from the point measurement data shown in Fig. 2E-F. Samples (or quantiles) of a given predictive distribution are generated by drawing samples from (or calculating quantiles for) the Gaussian predictive distribution described by the OK prediction mean and variance, and then applying the inverse normal score and detrending transformations as described in the prior two sections. The validation scheme for local (marginal) predictive distributions is described in Section 2.3 and the results of the validation

study are presented in Section 3.1.

### 2.2.4 Sampling from the Joint Predictive Distribution using Conditional Gaussian Simulation

In addition to OK, we also demonstrate spatial uncertainty quantification using conditional Gaussian simulation (CGS). Samples generated by CGS are equally probable "realizations" of the underlying random field that produced the measurement data. For a stationary Gaussian random field, local predictive distributions (i.e., the marginal predictive distribution at a given

grid cell) generated by CGS are equivalent to the distribution defined by the kriging variance (see e.g., Goovaerts, 2001).

However, CGS produces realizations that are spatially coherent with respect to the model of spatial covariance (see Sect. 2.2.5 Variography), whereas gridded estimates produced by OK do not. As such, CGS can be used to express "spatial uncertainty," or spatial correlation in errors, as described by the joint predictive distribution over multiple grid cells.

We apply CGS using a method based on a decomposition of the conditional distribution covariance matrix, commonly referred to in geostatistics literature as the "LU method" of CGS (Alabert, 1987). The LU method samples from the exact multivariate Gaussian predictive distribution. For larger scale simulations, the LU method becomes computationally infeasible and other algorithms and approximations may be required. However, in this example, we are drawing samples for a small number of locations so the LU method works well. As with OK, inverse normal score and detrending transformations are applied to produce the final predictive distributions.

This study does not include a comprehensive assessment of spatial uncertainty quantification. Instead, we present a case study (Section 3.2) using data from two weather stations close to one another where we expect the Tmax values and prediction errors to be correlated. Quantitative results using $10^4$ samples from the joint and marginal predictive distributions are shown that illustrate how CGS describes spatial uncertainty.

**2.2.5 Variography**

Both OK and CGS rely on a model of spatial covariance to produce gridded Tmax estimates and prediction uncertainty. Traditionally in geostatistics, spatial variation is represented using a semi-variogram, which is a function that describes the decrease in correlation between two locations as the distance between them increases (see e.g., Olea, 1999; Goovaerts, 1997). In this study, variography is performed using functions from the GeoStats.jl library (Hoffimann, 2018). We model the theoretical semi-variogram using the pentaspherical function:

$$\gamma(h) = (s - n)\left[\left(\frac{15}{8}\left(\frac{h}{r}\right) - \frac{5}{4}\left(\frac{h}{r}\right)^3 + \frac{3}{8}\left(\frac{h}{r}\right)^5\right) \cdot 1_{(0,r)}(h) + 1_{[r,\infty)}(h)\right] + n \cdot 1_{(0,\infty)}(h)$$

( 2 )

where the variable $h$ is the lag distance and $s$, $n$, and $r$ are parameters estimated to fit the empirical semi-variogram representing the sill (value of $\gamma$ as $h \rightarrow \infty$), nugget (value of $\gamma$ as $h \rightarrow 0$), and range (roughly the value of $h$ where $\gamma$ "levels off"), respectively. $1_{(l,u)}(h)$ is an indicator function that is 1 if $l < h < u$ and 0 otherwise. The theoretical model is fit to the empirical semi-variogram by minimizing the sum of squared errors with equal weight given to each lag bin. Model fitting is performed using the detrended and normal scored data (the data in Figure 2E-F) for each day independently (i.e., there is a different semi-variogram model for each day). At shorter lag distances, the pentaspherical model produces semi-variances between that of exponential and spherical models, which have been used previously to interpolate near-surface air temperature (Cornes et al., 2018; Hudson and Wackernagel, 1994). Running the analysis using these different semi-variogram models produces, in aggregate, very similar validation statistics. The pentaspherical model produces better accuracy in the CGS case

study, and we use it throughout the analysis for consistency. Interested readers can reproduce all results and figures in this paper using different semi-variogram models by replacing the model type passed to the semi-variogram fitting function (see Code Availability). Examples of fitted semi-variograms are shown in Fig. S1 in Supplementary Information.

At present, we do not model or try to exploit temporal correlation in Tmax or prediction errors. This topic will be addressed in future work.

## 2.3 Validation of Local Predictive Distributions

We implement a validation scheme that evaluates the accuracy of the OK predictive distributions in quantifying local prediction uncertainty. Uncertainty is not intrinsic to the physical phenomenon and, as such, there is not an objective "correct" predictive
distribution. However, a collection of "valid" predictive distributions should produce statistics that reflect appropriate levels of confidence in aggregate. The main quantitative validation results in this study are for local (marginal) predictive distributions computed using OK. We present some quantitative results for a CGS case study, but we do not conduct a comprehensive evaluation of spatial uncertainty quantification.

To assess the validity of local predictive distributions, we use a strategy based on prediction intervals described by
Deutsch (1997). For each day of year, we perform leave-one-out (LOO) cross-validation with each weather station. The measurement at the left-out station is not used in trend modeling, variography, or estimation. Predictions are made for the point at the center of the grid cell containing the weather station that will be used for validation. Estimates of the multiples of 0.005 quantiles are produced for each predictive distribution, from which centered prediction intervals are calculated with $q$ coverage for $q \in Q = \{0.01, 0.02, ..., 0.99\}$. For each $q$ prediction interval, let $q_{low} = (1 - q)/2$ and $q_{upp} = (1 + q)/2$ be the lower
and upper quantiles bounding the theoretical prediction interval. Then for $Tmax_i$, the measured Tmax at weather station $i$, define an indicator function:

$$\xi(Tmax_i ; q) = \begin{cases} 1, \text{if } q^*(Tmax_i) \in (q_{low}, q_{upp}) \\ \quad 0, \text{otherwise} \end{cases}$$

( 3 )


Where $q^*(Tmax_i)$ denotes the quantile of the of the true $Tmax_i$ relative to the predictive distribution for location $i$. Then for each $q$ prediction interval, we compute an average over all $n$ stations as $\overline{\xi(q)} = \frac{1}{n} \sum_{i=1}^{n} \xi(Tmax_i ; q)$. For exactly valid predictive distributions, $\overline{\xi(q)} = q$ for any $q$. To summarize the errors, we calculate the mean bias error as $\frac{1}{99} \sum_{q \in Q} \overline{\xi(q)} - q$ and the mean absolute error (MAE) as $\frac{1}{99} \sum_{q \in Q} |\overline{\xi(q)} - q|$. The MAE weights errors from being too confident and too
conservative equally. The bias indicates whether the predictive distributions are too confident (negative) or too conservative (positive) on average.

Figure 3 shows a representation of how the $\xi$ function is evaluated. The blue and red vertical lines denote the lower and upper bounds, respectively, of nested prediction intervals. The translucent lines indicate intervals that do not contain the true measured value, which is denoted by the vertical dashed line. The solid lines indicate that the interval does contain the true value. The prediction intervals are determined by the predictive distribution, the PDF of which is shown as a dotted curve.

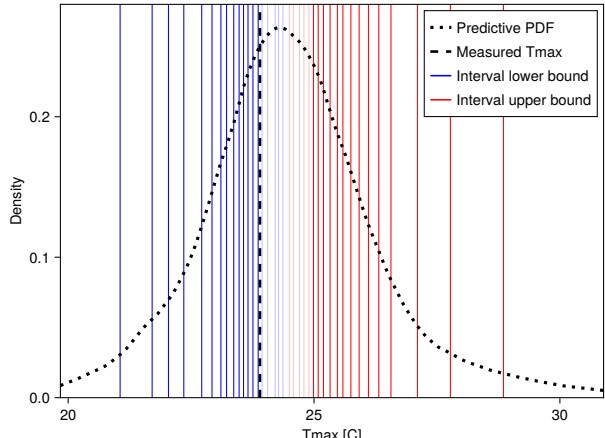

**Figure 3:** Prediction intervals versus measured Tmax. The lower bounds (blue) and upper bounds (red) of prediction intervals with coverage $q \in \{0.05, 0.10, ..., 0.95\}$ for a single prediction location are shown. (Validation statistics are computed using $q \in \{0.01, 0.02, ..., 0.99\}$, but we show fewer intervals here for legibility.) Intervals that contain the measured Tmax (black dashed line) are drawn as solid lines and the intervals that do not are drawn as partially transparent. The dotted black curve shows a kernel density estimate of the predictive distribution.

## 3 Results

### 3.1 Local Uncertainty Quantification using Ordinary Kriging

We first present the validation of local uncertainty quantification using the predictive distributions from OK. Figure 4 shows the average accuracy of the predictive distributions using the LOO validation scheme described in in Sect. 2.5. Figure 4A shows the mean absolute error (MAE) in terms of the predicted versus actual proportion of Tmax values that fall within a given prediction interval. For example, an MAE of 0.01 indicates that for a $p\%$ prediction interval (where $p = q \times 100$), the true value fell within that interval $(p \pm 1)\%$ of the time. The largest MAE of 0.045 occurs on DOY 96. The median MAE is 0.013 and 82% of days had an MAE less than 0.02. Fig. 4B shows the average bias for each day. For example, a bias of -0.01 indicates that for a $p\%$ prediction interval, the true value fell within that prediction interval $(p - 1)\%$ of the time. Positive bias indicates that the prediction intervals are too conservative on average, and negative bias indicates that the prediction intervals are too confident on average. The largest positive bias of +0.045 occurred on DOY 96 and the largest negative bias of -0.040 occurred on DOY 7. The median bias was -0.005. There are temporal patterns with the bias metrics errors of similar sign and magnitude persisting for periods ranging from a few days to multiple weeks.

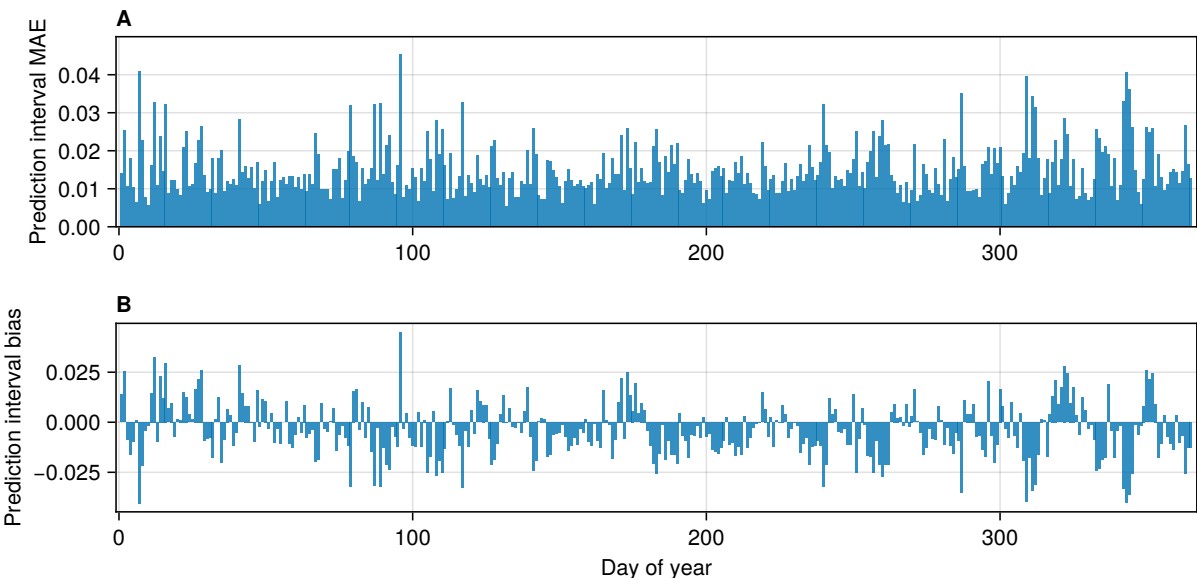

**Figure 4:** Predictive distribution quantitative validation statistics. Validity is assessed by comparing the theoretical coverage versus actual rate that measurement data fall within prediction intervals. The calculation uses an indicator function $\xi$ described in Eq. (1). Panel A shows the mean absolute error (MAE) for each day of year, calculated as $\frac{1}{99}\sum_{p=1}^{99}|\overline{\xi(q)} - q|$. Panel B shows the mean bias error for each day of year, $\frac{1}{99}\sum_{p=1}^{99}\overline{\xi(q)} - q$. A positive bias indicates that predictive distributions were too conservative on average and negative bias indicates they were too confident on average.

Figure 5 shows validation results for three individual days. In addition to showing the day with the median MAE, we highlight the two days with the most significant errors in the sample to give a sense of the "worst case" accuracy in uncertainty quantification. Values in the x-direction correspond to theoretical prediction intervals centered on the median. Values in the y-direction are the actual proportion of true Tmax values that fall within the given theoretical interval. For example, a point at (0.25, 0.3) would mean that 30% of true values fell within the corresponding 25% prediction intervals. Figure 5A shows results for DOY 304, which had the median MAE of 0.013. For DOY 304, the actual proportions closely track the theoretical prediction intervals with the largest error occurring at the 74% intervals, which contained 77.1% of the true values. Figures 5B and 5C show the same information for DOY 96 and 7, which are the days with largest positive and negative biases of +0.045 and -0.040, respectively. The largest error for DOY 96 occurs at the 60% prediction intervals, which contained 68.1% of true Tmax values. The largest error for DOY 7 occurs at the 62% prediction intervals, which contained 54.9% of true Tmax values.

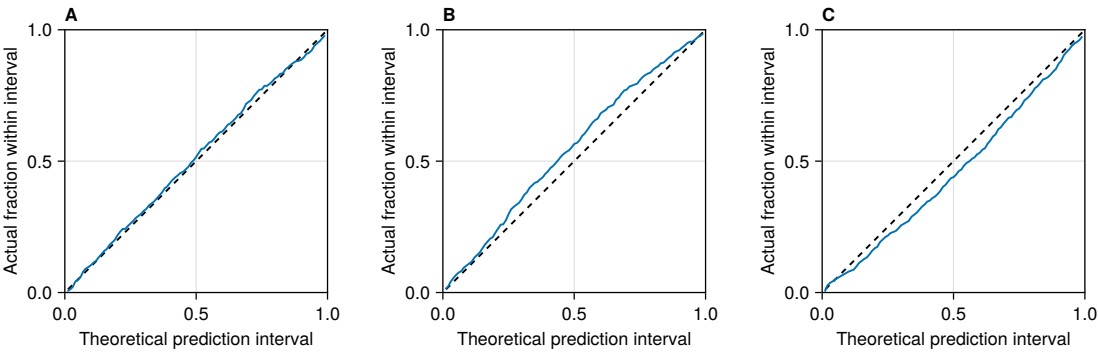

**Figure 5:** Theoretical versus actual coverage of prediction intervals. X-axes correspond to the coverage of the theoretical prediction interval and Y-axes correspond to the proportion of values that actually fell within the intervals. Panel A shows results for DOY 77, which had the median MAE of all the days in the sample. Panels B and C shows DOY 96 and 343, which had the largest positive and negative biases, respectively.

When applying our method across a full spatial domain, we observe that local predictive distributions vary in space and time. Figure 6 shows predicted Tmax and uncertainty statistics (spread of predictive distributions) for four different days, where each row corresponds to a day. The days are approximately equally spaced and cover different seasons, including the first days of January (Fig. 6A-6C), April (Fig. 6D-6F), July (Fig. 6G-6I), and October (Fig. 6J-6L). The first column (Fig. 6A, 6D, 6G, and 6J) shows the median of the predictive distribution for each 1 km² grid cell. The second column (Fig. 6B, 6E, 6H, and 6K) shows the magnitude of the 50% prediction interval of each predictive distribution (the 75th percentile minus the 25th percentile). The third column (Fig. 6C, 6F, 6I, and 6L) shows the magnitude of the 90% prediction interval of each predictive distribution (the 95th percentile minus the 5th percentile). Each column uses a single common color gradient (the gradients do not vary between rows).

The Tmax prediction uncertainty varies in both space and time. Comparing the uncertainty maps in time (between rows), the magnitude, spatial patterns, and magnitude of variation in prediction uncertainty all vary. Spatial patterns also differ based on the magnitude of the prediction interval. OK prediction variance depends only on the distances to measurement data locations, not the measurement values themselves (see e.g., Olea, 1999). However, the forward and inverse normal score transforms are nonlinear, so the shape of the final prediction distribution is influenced by the actual measurement and prediction values. The 90% prediction intervals (rightmost column) appear to be controlled primarily by the local spatial density of measurements, with the locations of weather stations standing out as local minima. The patterns in the 50% prediction intervals (center column) are more complex. The fact that many of the locations with the largest uncertainties are near the Pacific coast may be caused by the trend model failing to capture local Tmax trends, where cooling from the ocean confounds the usual negative correlation between temperature and elevation. This could cause the detrended measurements near the coast to be outliers on the low end of the overall distribution. As a result, prediction intervals in this part of the distribution get "stretched out" by the inverse normal score transform. This issue would not occur in the mountains where the expected relationship

between temperature and elevation holds, and the resulting detrended data are not outliers. A more sophisticated approach to trend modeling may improve accuracy and reduce uncertainty near the coast.

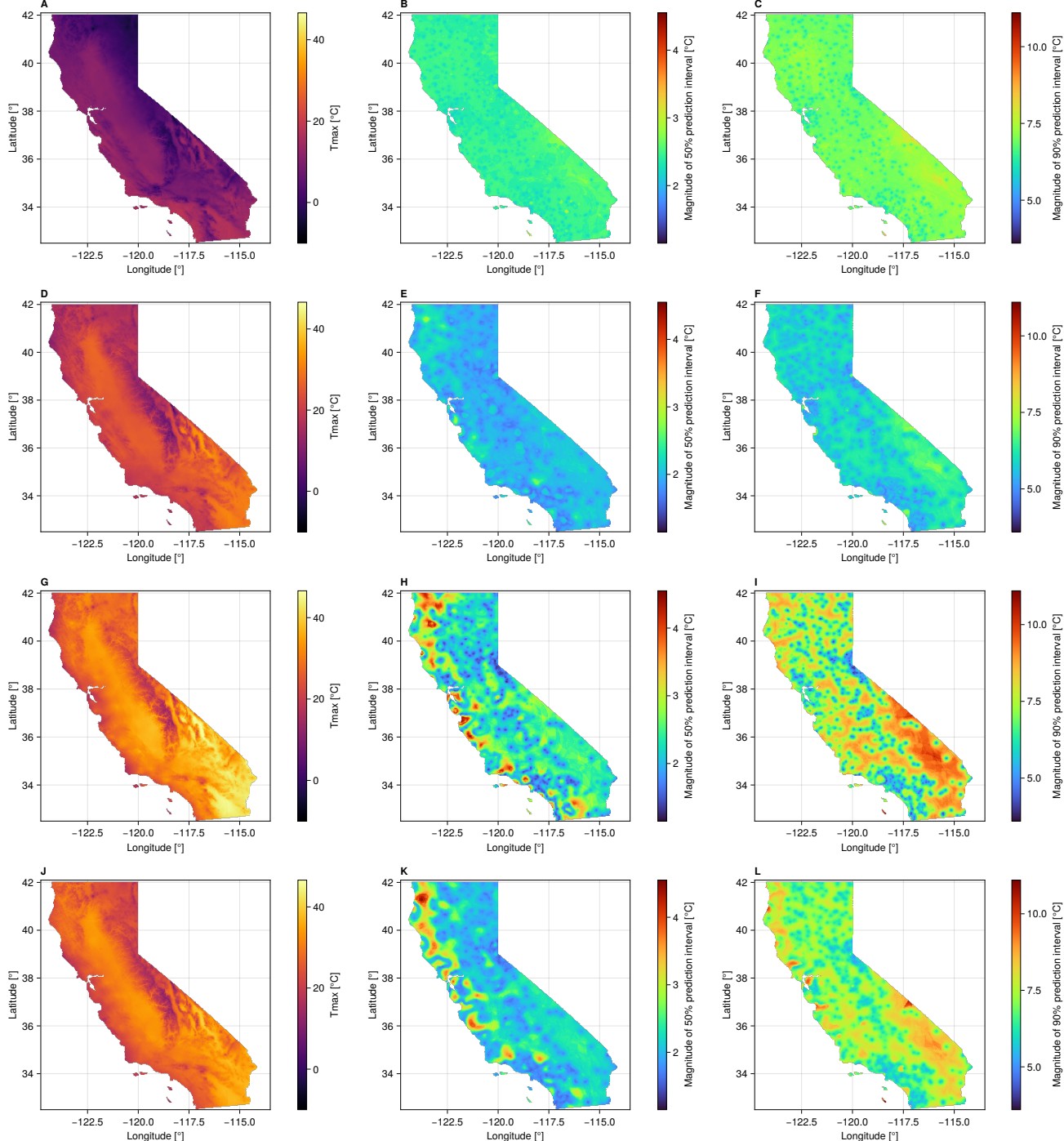

**Figure 6:** Tmax and prediction uncertainty maps for four different days. Each row shows results for a different day. The days are DOY 1 (panels A-C), DOY 91 (panels D-F), DOY 182 (panels G-I), and DOY 274 (panels J-L) from 2022. The first column (panels A, D, G, J) shows the median of the Tmax predictive distribution at each grid cell. The second column (panels B, E, H, K) shows the magnitude of the 50% prediction interval for each grid cell. The third column (panels C, F, I, L) shows the magnitude of the 90% prediction interval for each grid cell.

## 3.2 Case Study: Spatial Uncertainty Quantification using Conditional Gaussian Simulation

In addition to local (cell-wise marginal) uncertainty, we can represent spatial uncertainty using CGS to sample from the joint predictive distribution over multiple grid cells. Figure 7 shows the predictive distributions for grid cells containing two weather stations, IDs USC00043747 ("station 1") and USW00053119 ("station 2"), near Hanford, California for DOY 182. The centers of the grid cells containing the stations are 1.4 km apart. The red lines show kernel density estimates of the joint and marginal predictive distributions generated using CGS. The blue lines show the same predictive distributions generated using OK, where the "joint distribution" is generated by sampling independently from the marginal distributions. The empirical distributions were generated using $10^4$ samples each. While the OK samples show no correlation between locations (by construction), the samples produced by CGS show a correlation of approximately 0.66 between predictions at the two locations. This shows how errors in predictions at nearby locations are likely to be correlated with one another. The marginal distributions generated using OK and CGS are virtually identical at each of the two locations (Fig. 7A and 7C), as expected.

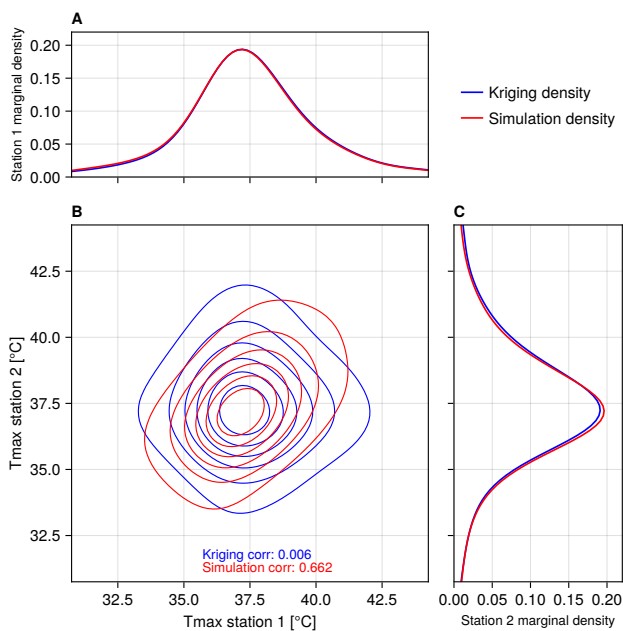

**Figure 7:** Joint and marginal predictive distributions using OK versus CGS. Predictive distributions were generated for two nearby grid cells that contain weather stations. Panels A and C show the marginal (local) predictive distributions at the two locations and panel B shows the

joint (spatial) predictive distribution. The predictive distributions from OK are in blue and the distributions generated by CGS are in red. The two estimation methods produce the same marginal distributions, but the joint distributions are different. The distribution from CGS reflects the spatial correlation in predictions between the two locations.

Figure 8 shows histograms of predictive distributions for Tmax estimates at Station 1 when conditioned on different information about Tmax at Station 2. Measurements from both stations are held out from the estimation procedure. The dashed black line is the Tmax measured at Station 1. The blue histogram is the unconditional marginal distribution for the grid cell containing Station 1. The orange and green histograms show empirical conditional distributions at Station 1 given that the error at Station 2 is less that 2 degrees ("conditional 2") and less than 1 degree ("conditional 1"), respectively. These empirical

distributions are generated by filtering the simulation ensemble. Conditioning the predictive distribution reduces the standard deviation from 2.4 °C in the unconditional distribution to 1.9 °C and 1.7 °C for conditional 2 and conditional 1, respectively. If the mean of the predictive distribution is taken as a point estimate, the error for the unconditional distribution is +0.25 °C. The error is reduced to -0.08 for conditional 2 but then increases (in magnitude) to -0.25 for conditional 1. The degree to which the point estimates and prediction variance change is controlled by the semi-variogram model. The semi-variogram for DOY

182 has the smallest nugget (i.e., the strongest correlation at short distances) among the days shown in Fig. 6. This means the model-implied correlation between the two stations will be lower on the other three days. Plots of empirical and fitted theoretical semi-variograms for the four days can be found in the Supplementary Information. In general, we observe that the fitted semi-variograms have relatively large nugget values. This may be partly an artifact of the semi-variogram estimation process, given that there are few pairs of weather stations within 1-10 km of one another. However, there does appear to be

significant "real" variance in measured Tmax values even at short spatial scales. For example, the two stations used in this example are 1.4 km apart and at nearly identical elevation but have an average difference in Tmax of nearly 1.5 °C in 2022. This difference may be explained by site-specific effects, such as land cover or other features near the weather stations, rather than physical variation that would persist under idealized conditions.

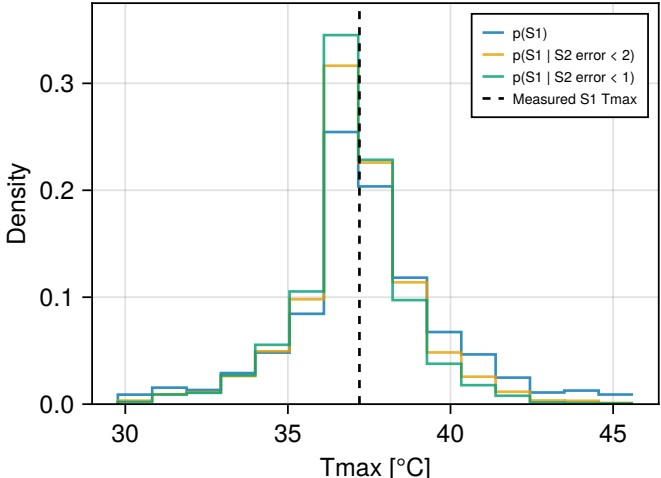

Figure 8: Marginal predictive distributions without and with additional spatial information. Three empirical predictive distributions at Station 1 (S1) are shown including the unconditional (blue) and two distributions conditioned on predictive accuracy at the nearby Station 2 (S2). Conditioning is performed by filtering the ensemble and leaving only realizations where the prediction at S2 was within 2 °C (orange) or within 1 °C (green) of the measured value. Conditioning on additional spatial information reduces the spread of the marginal predictive distribution.

## 4. Discussion and Conclusion

This work presents and validates an approach to quantifying spatially and temporally resolved prediction uncertainty in interpolated meteorological data products. In the quantitative validation study, the DNK method produced highly accurate uncertainty quantification. Even in the least accurate cases, the predictive distributions are qualitatively informative and are still sufficiently accurate to be useful for many applications. Accuracy assessment of point estimates is useful but fundamentally cannot describe prediction uncertainty at times and locations where measurement data are unavailable. In this application, we have measurements at hundreds of weather stations and predict Tmax values at hundreds of thousands of grid cells, some of which are nearly 100 km from the nearest measurement. This means that the number of locations where we can assess prediction uncertainty using actual measurement data is vanishingly small compared to the number of locations where we do not have measurements. We show that the magnitude of prediction uncertainty varies significantly in space and in time, and that using average error statistics will overestimate prediction uncertainty in some cases and dramatically underestimate it in others.

The DNK methodology described in this paper could be applied to a wide range of applications due to its generality and relatively low computational cost. The main application area motivating this work was modeling of land surface fluxes of water and carbon, given that rates of evapotranspiration (Volk et al., 2024) and primary production (Zeng et al., 2020) are particularly sensitive to near-surface meteorological conditions. However, gridded meteorological data are used to make

predictions and draw conclusions about many other phenomena including crop yield (Lobell et al., 2015), vegetation phenology (White et al., 1997), economic productivity (Burke et al., 2015), human conflict (Hsiang et al., 2013), and others. Using gridded predictive distributions rather than point estimates can help ensure the robustness of scientific conclusions given uncertainty in model inputs. The decision to use cell-wise marginal (OK) or full joint (CGS) predictive distributions depends on various factors including the size of grid cells, the distances between locations being compared, and sensitivity of the analysis to spatial correlation in prediction errors (e.g., for causal inference). Users of gridded meteorological data products can propagate uncertainty through their analysis by running models multiple times using either random samples or a preset collection of quantiles from the distribution. This approach does not require any additional modeling choices or assumptions because the relevant information about uncertainty in the model inputs is expressed by the predictive distributions. Also, computationally expensive models may use the gridded variable uncertainty to prioritize sensitivity analyses and reduce the total number of model runs required.

Producing a predictive distribution using DNK, rather than a single point estimate, requires only marginally more computation. The main computation required in OK is solving for the kriging weights $\lambda_s$, which requires solving a $(S + 1) \times (S + 1)$ system of linear questions for $S$ stations. The prediction mean $\mu_{OK}$ and variance $\sigma_{OK}^2$ are both functions of the $S$ station data and the weights $\lambda_s$. Drawing samples from the predictive distribution only requires drawing samples from Normal($\mu_{OK}, \sigma_{OK}^2$) and then, for each sample, applying the reverse normal score transformation and adding back the trend. Computing the joint conditional distribution and sampling using CGS is more computationally expensive, as it requires solving the system of kriging equations as well as factorizing a $C \times C$ matrix, where $C$ is the number of prediction grid cells and often $C \gg S$. There exist more computationally scalable methods for conditional Gaussian simulation (Gómez-Hernández and Journel, 1993; Gutjahr et al., 1997; Gómez-Hernández and Srivastava, 2021) that we do not discuss here.

The normal score transformation used in this study is an important step that may explain differences in our validation results compared to prior studies. Cornes et al. (2018) evaluated the predictive distributions of the E-OBS data product using a similar approach to the one used in this work. They found that, on average, the predictive distributions were over-confident and that measured values fell in the tails of the distributions at a higher-than-expected rate, although this trend varied regionally. Our study is conducted over a much smaller area and with different prediction spatial scales (1 km versus 12 km), so we cannot definitively explain the differences in findings. However, omission of the normal score transformation could explain under-dispersion in predictive distributions. We observe that the distributions of the detrended observation data generally have positive excess kurtosis, meaning they have heavier tails than a Gaussian distribution. As such, approximating predictive distributions with a Gaussian could fail to account for additional probability mass in the tails. Surrounding the calculation of the predictive distributions with forward and inverse normal score transforms does a better job of capturing the shape of the original data distribution.

Our analysis also provides perspective on when the additional computational cost of CGS is justified. Cornes et al. (2018) note the high cost of CGS, including the fact that it was prohibitive for earlier versions of the data product. In general, CGS is required when spatial correlation in errors matters for the application, like prediction uncertainty for spatial averages.

However, when examining a single location in isolation, the predictive distribution given by OK is valid and CGS does not add value. The semi-variogram model, and the nugget in particular, determine the degree of correlation at shorter spatial scales. For distances beyond some correlation length, the joint distribution can be reasonably approximated as the product of the marginals. While air temperature is traditionally modeled as varying smoothly in space with semi-variogram nuggets close (Cornes et al., 2018) or equal to zero (Hudson and Wackernagel, 1994), observational Tmax data do not necessarily support

this assumption (see Section 3.2). This topic merits further inquiry.

Users of the DNK methodology should be aware of some practical limitations. First, we reiterate that there is no objectively correct Tmax predictive distribution for a given location. Uncertainty is a property of the measurement data and modeling decisions, and making different modeling decisions will produce different predictive distributions. Different modeling decisions could lead to larger or smaller errors in point estimates on average, but still produce predictive distributions

that are valid (i.e., where true values fall in prediction intervals at the prescribed rate). In addition, even large samples from predictive distributions will necessarily suffer from deficiencies inherent in the estimation and sampling process. Covariance stationarity and multi-Gaussianity are strong assumptions that are relied upon for the validity of the predictive distributions, and the transformations made to satisfy these assumptions are imperfect. The normal score transformation requires estimating an empirical distribution from the weather station data. Given that we generally wish to draw samples larger than the size of

the measurement data, reverting the normal score transformation necessarily requires interpolation and extrapolation of that distribution (see Goovaerts (1997) for a detailed discussion). In practice, this can produce artifacts like clusters of similar values, particularly near under-sampled portions of the distribution. Regarding stationarity, trend modeling can also strongly influence predictive distributions. Like uncertainty itself, a "spatial trend" is not an objectively observable phenomenon. Reliably estimating spatial trends can be difficult when measurement data are sparse or when other physical phenomena, like

cooling or warming due to coastal proximity, confound the basic estimation procedure. Limitations in trend modeling contributed to our use of OK (locally constant unknown mean) rather than Simple Kriging (SK) (locally constant known mean), despite the latter being theoretically justifiable for detrended data. Results using OK versus SK were very similar, with OK performing marginally better likely due to coarse scale variation that was still present after modeling and subtracting spatial trends. Finally, note that the validation results for this study were produced using high quality Tmax data from GHCN. If the

data contain non-random errors, the assumptions of the DNK procedure will no longer be satisfied and the quality of uncertainty quantification may decline. For less densely sampled study areas, prediction uncertainty will grow for many locations, but the predictive distributions should still be valid. It would be valuable to evaluate the DNK methodology using poor quality and sparse but doing so would require a different experimental design and is beyond the scope of this study.

There are many potential avenues for future work building on the methods and results described in this paper. One

important area for further study is the analyzing the effects of trend estimation on characteristics and robustness of predictive distributions. For a covariance stationary and multi-Gaussian random field, predictive distributions will be valid over a sufficiently large sample. This indicates that invalid predictive distributions are driven primarily by the transformations we apply (and revert) to make the data stationary and Gaussian. Relatedly, it would be useful to find ways of incorporating

additional physical information not explained by a large-scale spatial trend. The strength of data products like NEX-GDM and PRISM come from the use of additional physical information (e.g., coastal proximity, slope, aspect) in predictions. It is not immediately clear how this information could be incorporated into the underlying mathematical model from which our predictive distributions are derived, but doing so could produce more precise estimates accompanied by valid predictive distributions. Lastly, the DNK method should be tested for interpolation of meteorological variables other than Tmax. The methodology seems likely to transfer to certain variables like daily minimum air temperature and humidity, but it is not a given that all variables can be approached the same way.

Going forward, it will be valuable to not only produce the "best available" gridded meteorological data products, but also to produce spatially and temporally resolved uncertainty quantification. Accounting for uncertainty in model inputs is important considering the robustness of scientific conclusions. It is also important for guiding the design and implementation of science-informed policies. Given uncertain information, conclusions about the "best" policy option may differ when using a deterministic versus probabilistic benefit-cost analysis (Morgan and Henrion, 1990). Similarly, there may be asymmetric consequences for over- or underestimation of a given model input. In this scenario, using a predictive distribution rather than a point estimate allows policymakers to quantitatively assess tradeoffs between maximizing expected outcomes and minimizing risk. More broadly, accounting for uncertainty in scientific models is necessary not only for designing informed policies, but for building and maintaining trust in science-informed policymaking.

**Code Availability**

All code to perform analysis and generate figures is available on Zenodo at https://zenodo.org/doi/10.5281/zenodo.12171025 and on GitHub at https://github.com/conordoherty/met-uncertainty-paper.

**Data Availability**

Tmax data from Thornton et al. (2022) are available for download from the ORNL DAAC at https://doi.org/10.3334/ORNLDAAC/2132. All other required data are included in the code repository and can be downloaded from Zenodo or GitHub.

**Author Contributions**

CD and IB developed the concept and acquired funding. CD developed the methodology, developed the software, performed the experiments and formal analysis, and wrote the initial draft of the manuscript. WW contributed to methodology development and formal analysis. HH contributed to methodology development, validation, and software testing. All authors contributed to reviewing and editing the manuscript.

**Competing Interests**

The authors declare that they have no conflict of interest.

**Acknowledgements**

CD is supported by an appointment to the NASA Postdoctoral Program at NASA Ames Research Center, administered by Oak Ridge Associated Universities under contract with NASA. CD thanks Forrest Melton for support on the initial conceptualization and proposal for this work, A.J. Purdy and Lee Johnson for helpful feedback on the manuscript, and Kyle Kabasares for performing additional software testing for reproducibility. CD also thanks Júlio Hoffimann for his generous assistance in using the GeoStats.jl library, which he created.

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
