# Peer review of "A Method for Quantifying Uncertainty in Spatially Interpolated Meteorological Data with Application to Daily Maximum Air Temperature"

_EGUsphere, 2024_

## Author Response (AR1)

**Referee comment**
Initial response
Updated response
Excerpt of manuscript revision

Dear Editor and Referees,

In addition to what is described in line with the referee comments below, we made the following revisions to the manuscript:

1. While investigating a question posed by Referee 3, we discovered that the elevation metadata associated with two weather stations was incorrect (the stations were listed as being at sea level, but clearly are not). For these stations, we are now using the elevation from the DEM in the trend estimation. This caused the validation statistics and figures to change very minorly. However, there is no substantive difference in our analysis or conclusions. (We also reported this error to NOAA NCEI, who supply the GHCN data.)
2. Figures 7 and 8 are now showing predictive distributions for DOY 182. Of the four days shown in Fig. 6, DOY 182 is the one with the strongest correlation at fine spatial scales.

All other non-cosmetic changes were made in response to referee comments, which are documented below.

**Referee 1**

We thank the referee for their time and attention. We particularly appreciate the referee identifying relevant prior work that we missed in the literature review and their overall attention to detail. Addressing the referee's comments will improve the quality of the paper.

**A critical point of this manuscript is that the conclusion is based only on a single year (2022). It would be, especially in context to climate-impact modelling, interesting to see how the method works in different years (e.g. in earlier years where data with lower data quality is more present) and if there is an influence on the uncertainty distributions.**

The reviewer raises an important question, the answer to which will affect when and how the method can be applied. The purpose of this paper is to describe and validate the method, which we do using high quality data. Including additional years, when also using data that are highly available and believed to be accurate, would produce results that are qualitatively similar to what we already include and would be essentially redundant. However, the referee's comment seems to be more focused on validation using lower quality data. This is a valid question, but one that we believe merits its own future study.

Inferior data quality must be addressed in the pre- and post-processing data transformations, not the estimation (kriging/conditional Gaussian simulation). If the measurement data can be transformed to satisfy the assumptions of a stationary Gaussian random field, then the prediction intervals will still be valid. If the data are spatially sparse, the uncertainty will increase at most prediction locations. Even if the data contain noise (i.e., random error, not bias*), the estimation can accommodate this by modifying the semivariogram accordingly. However, in practice, transforming the data to satisfy these assumptions will likely be more difficult. Addressing this question would require an in depth analysis of trend modeling methods with respect to errors in measurement data and spatially preferential sampling. Given the requirements for exposition and/or development of additional methods, as well as the need for new experimental design, we believe this analysis should be done in a separate future work.

We will add some of this content to the Discussion section, as some readers will likely have the same question.

* Dealing with biased measurement data is yet another issue that would require fundamental changes to the approach.

**Line 424**: Finally, note that the validation results for this study were produced using high quality Tmax data from GHCN. If the data contain non-random errors, the assumptions of the DNK procedure will no longer be satisfied and the quality of uncertainty quantification may decline. For less densely sampled study areas, prediction uncertainty will grow for many locations, but the predictive distributions should still be valid. It would be valuable to evaluate the DNK methodology using poor quality and sparse but doing so would require a different experimental design and is beyond the scope of this study.

**It was mentioned that grid points close to the Pacific coast have higher uncertainties than others. It would be interesting how the method works in complex terrain, i.e. the high-elevated areas of California. From Figure 6 I assume that high-elevated areas have smaller uncertainties than coastal areas. Maybe the authors can include some information on that in the text.**

~~"We hypothesize that the larger uncertainties near the coast are caused by two distinct but related phenomena. First, complex interactions between ocean-driven cooling and near-coast topography lead to significant variability over fine spatial scales. Second, the local linear trend model is not well-suited to capture the effects of ocean-driven cooling, which can confound the usual relationship between temperature and elevation. In estimating and "removing" the trend, it may artificially increase local variability, which then leads to larger variance in the predictive distribution. Both sources of variability (real and artificial) are induced by the effects of the ocean, which is why change in elevation alone does not necessarily increase prediction~~

The following explanation is more thorough and precise than what is written above:

Line 288: OK prediction variance depends only on the distances to measurement data locations, not the measurement values themselves (see e.g., Olea, 1999). However, the forward and inverse normal score transforms are nonlinear, so the shape of the final prediction distribution is influenced by the actual measurement and prediction values. The 90% prediction intervals (rightmost column) appear to be controlled primarily by the local spatial density of measurements, with the locations of weather stations standing out as local minima. The patterns in the 50% prediction intervals (center column) are more complex. The fact that many of the locations with the largest uncertainties are near the Pacific coast may be caused by the trend model failing to capture local Tmax trends, where cooling from the ocean confounds the usual negative correlation between temperature and elevation. This could cause the detrended measurements near the coast to be outliers on the low end of the overall distribution. As a result, prediction intervals in this part of the distribution get "stretched out" by the inverse normal score transform. This issue would not occur in the mountains where the expected relationship between temperature and elevation holds, and the resulting detrended data are not outliers. A more sophisticated approach to trend modeling may improve accuracy and reduce uncertainty near the coast.

**Other gridded observation-based datasets have already uncertainty estimations by calculating ensembles, for example the European E-OBS dataset (https://surfobs.climate.copernicus.eu/userguidance/use_ensembles.php). It would improve the overview of the current research status by providing this information. Cornes et al. (2018): An Ensemble Version of the E-OBS Temperature and Precipitation Data Sets, https://doi.org/10.1029/2017JD028200**

Thank you for bringing this work to our attention. There are many similarities with our work, but some important differences as well. Results from our work also provide context for practical and technical matters discussed in this paper. We will revise the Discussion in our manuscript to explain all of this in detail, but briefly summarize a few points here:

- Cornes et al. assume that the raw weather data (or the square root thereof for precipitation) are Gaussian. They do not apply (or at least do not mention) a normal score transformation (the N in DNK). While the kriging estimator gives the prediction variance, if the data are not Gaussian, then the predictive distribution is not fully characterized by the mean and variance. Relatedly, the quantiles of the "true" predictive distribution will not necessarily correspond to those of a normal distribution with the same moments. This may explain the discrepancies between the expected and the actual coverage of the predictive distributions (their Fig. 3).
- The authors note that, in previous iterations of the work, full realizations of spatially correlated random fields were not produced due to the computational cost. However, these ensembles were produced starting with version v16.0e described in the 2018

paper. Our work provides additional information about when and why full realizations are needed versus when marginal predictive distributions (using kriging) will suffice. In the case of Tmax estimation for our study region, we find that the additional computation adds relatively little value. We will expand the explanation of this topic in the Discussion.

Cornes et al. (2018) is a very relevant to this work. We add a reference to it in the introduction, and then there is a more technical discussion of difference in the methods/results starting on line 386 of the revised manuscript.

**The abstract needs some restructuring. The information is provided but it would be easier to read if the authors could structure it following recommended guidelines (motivation, task/method, findings, conclusion, and perspective).**

We will revise the abstract (per suggestion of all three referees). Done.

**Chapter 2.1. Input Data: It would be good to mention the analysed year (2022) in this chapter already.**

We will add this. Line 88: We use data from 2022 to conduct the validation study.

**Figure 3: This figure is not referenced in the next. The authors are asked to do that. Furthermore, it is not completely clear to me, why intervals close to the measured Tmax value are transparent (do not contain the measured Tmax) while intervals further away do so. Maybe the authors can comment on that.**

The explanation above was incorrect – it is actually the reverse (solid lines denote intervals that contain the true value, semi-transparent do not). The correct explanation was given in the caption to Fig. 3. It is now also included in the body of the manuscript.

Line 232: Figure 3 shows a representation of how the $\xi$ function is evaluated. The blue and red vertical lines denote the lower and upper bounds, respectively, of nested prediction intervals. The translucent lines indicate intervals that do not contain the true measured value, which is denoted by the vertical dashed line. The solid lines indicate that the interval does contain the true value. The prediction intervals are determined by the predictive distribution, the PDF of which is shown as a dotted curve.

**L74: 2 m (with whitespace between number and unit) instead of 2m.**

Fixed.

**L224: It should be first day (instead of days) of January.**

I believe it should be "days" because it refers to multiple days (the first of each of four months), but I am open to editorial.

**L249: In the text the authors use station but in the related figure location. It would be nice to use only station or location.**

Fixed.

**L270: In the available version of the manuscript, it is a dashed black line (not a dotted). Maybe the authors can check.**

Fixed.

**L274: The authors write 2.0 C. I assume they mean 2.0 degree Celsius. To avoid any misinterpretation I recommend writing 2.0 degree Celsius instead of 2.0 C. The authors should also check other given absolute temperature values.**

Fixed.

**Referee 2**

We thank the referee for their time and attention. We particularly appreciate the referee carefully reviewing and catching errors in the equations in section 2.5. (Fortunately the errors are in how the equations were written and not in how the validation was actually performed; see comments below.) The referee also offered multiple good suggestions regarding the structure and organization of the paper.

**I think that it would be worth including Bayesian approaches in the literature review in the introduction (lines 43-58). They produce posterior predictive distributions, similar to what the authors want to achieve. For geostatistical data this can be done via SPDE. See example in meteorology "Interpolating climate variables by using INLA and the SPDE approach" by Fioravanti et al.**

Thank you for providing the reference. We will address this in the introduction. Unfortunately that paper does not include validation of the coverage of the predictive distributions. That would be interesting to see and compare.

Line 54: Finally, Bayesian methods using Gaussian Markov Random Fields (GMRF) have also been used to perform probabilistic interpolation of meteorological data. For example, Fioravanti et al. (2023) applied these methods to air temperature (but did not quantitatively validate the uncertainty quantification) and Ingebrigtsen et al. (2014) applied them to precipitation data. In future work, it would be instructive to compare predictive distributions produced using QRF and

Bayesian GMRF-based methods with those produced using the more classical geostatistical methods described in this work.

**While the acronym "DNK" is used in the abstract and introduction, in the methods section we don't find it until section 2.3, where it is assumed that the reader knows what it is. Moreover, the order of the methods sections does not correspond to the order of the steps taken in DNK, which I find a bit confusing. To address this, I suggest presenting an overview of DNK with a a numbered bullet point list where the steps are listed sequentially, and then explain in detail the different steps in order.**

The list is a good idea. We will add it at the beginning of the methods section.

We reorganized the methods section and added a brief overview and list of steps.

**Section 2.2: I think that the methods regarding simulation with OK could be better explained. This could be achieved by simply moving information included in the discussion (lines 319-323) to the methods section, especially how the simulations are performed via a normal distribution using the OK-estimated mu and sigma for each location. Likewise, details of CGS (lines 324-325) could also be moved to the methods.**

We will revise this section per the suggestion of referees 2 and 3.

This should be clearer in the revised Methods section.

**Section 2.2: I don't see the number of realizations you simulated to compute the predictive distributions and I think this information should be clearly stated.**

We will add text clarifying this. For the CV with kriging, we do not actually draw samples; we just compute the quantiles of the standard normal and apply the back transformations (the transformations preserve ordering). For the comparison of kriging and CGS, we generate 10000 samples.

The validation for OK uses quantiles as described in section 2.3. For CGS, line 182: Quantitative results using $10^4$ samples from the joint and marginal predictive distributions are shown that illustrate how CGS describes spatial uncertainty.

**Section 2.4 I think that the unfamiliar reader would benefit from a graphical representation, even in the appendix, of one the fitted semivariograms. This might be more helpful than the included formula to get a grasp of the discussed differences between exponential, spherical, and pentaspherical. Also later in the results section you**

**refer to the estimated nugget values (lines 278-280) but we have never seen the fitted semivariogram nor know the value of the estimated parameters.**

This is a valid point. We will add an appendix with some example variograms and briefly demonstrate the sensitivity to the semivariogram parameters. While air temperature in general is usually assumed to be highly spatially continuous, Tmax appears to be less so. This may be due to differences in weather station siting or other random fluctuations.

The updated submission includes a SI file with a plot of semi-variograms for the four days shown in Fig. 6.

**Section 2.4 It is not clear to me whether the variogram is fitted every day or in the contrary a "pooled" variogram is used for all time points. I understand the whole DNK is fitted to each day independently (so semivariogram parameters change by DOY), but I think it's worth clarifying.**

We will clarify this. New theoretical semivariograms are computed for each day (as are the trend values).

Line 196: Model fitting is performed using the detrended and normal scored data (the data in Figure 2E-F) for each day independently (i.e., there is a different semi-variogram model for each day).

**Section 2.5: I think the definition of p_low and p_upp are incorrect when p is defined in the scale 0-100. In the original text by Deutsch, it does make sense because p is expressed in the scale 0-1. I suggest either changing the formulas to p_low=(100-p)/2, or use the original definition by Deutsch.**

Yes, we accidentally switched between integer percentiles and corresponding quantiles in the formulas. We will fix this. The values are computed correctly in the actual code, though.

We revised this section to use quantiles rather than percentiles in the equations.

**Section 2.5: Likewise, I think that the definition of the indicator function is incorrect. Right now, the authors are directly comparing the values of Tmax with the percentiles, which doesn't make sense. We need a function that maps Tmax to its percentile in the distribution; in the original work Deutsch uses a CCDF that does precisely that, I suggest using the same.**

Again you correctly point out an error in the text. However in the actual calculation we do what you/Deutsch describe. We will fix this in the text. (Note that we diverge from Deutsch in that we assign equal weight to "overconfidence" and "underconfidence," while Deutsch assigns a greater penalty to the former.)

We revised this section to use quantiles rather than percentiles in the equations.

**Figures 3-6: Since you have two methods to compute the prediction distributions, I think it would be worth including in the figure captions that results correspond to OK to avoid confusion.**

Per another of the referee's suggestions, we divided the Results into sub-sections describing local (OK) versus spatial (CGS) uncertainty. The estimation method for each result/figure should now be clear from context.

**There are (very interesting!) analyses in the results section that have been not described in the methods and hence they come a bit as a surprise. These include results in Figures 7 and 8. You could include a new methods subsection where you explain how and why you do them.**

Yes, we will revise that section of the methods per comment from you and referee 3.

The revised methods section contains sub-sections describing kriging and conditional simulation separately.

**I miss a discussion regarding temporal autocorrelation naturally present in the data, which we can observe in the biases in figure 4B. There is the possibility of using a spatiotemporal geostatistical model to account for it, which could be listed as a possible extension.**

We briefly mention on line 343, but it could be discussed more explicitly. However, we are not planning to add significant discussion of temporal autocorrelation because it is a large topic (which are currently investigating and drafting a manuscript; see https://meetingorganizer.copernicus.org/EGU24/EGU24-13593.html).

Line 205: At present, we do not model or try to exploit temporal correlation in Tmax or prediction errors. This topic will be addressed in future work.

**Abstract restructuring: I suggest moving sentences in lines 13-16 "Uncertainty is inherent in gridded meteorological data…" to the top to motivate the study and then move to the objectives, methods and analysis. Right now the abstract motivates the study, presents the objectives and methods, and then goes back to the motivation, which I find slightly confusing.**

We will revise the abstract (per suggestion of all three referees). Done.

**Line 39-40: "(geostatistical) methods are not used in the most popular near-surface meteorological data products." While this is true, there have been attempts to do it that might be worth mentioning, see https://doi.org/10.1016/j.cageo.2007.05.001**

Yes this statement could be more nuanced. We will revise.

Line 36: […] these methods are not as widely used in popular near-surface meteorological data products.

**Only as a suggestion, I feel the structure of the results section could be improved by the addition of subsections that link to the methods and objectives rather than one big section without separators.**

Maybe we will divide the CV kriging results from the CGS versus kriging analysis.

We divided Results into to sub-sections, per the referee's suggestion.

**Also as a suggestion, in the field of spatial ML, other feature-based methods have been proposed to estimate spatially-explicit uncertainty, which might be worth mentioning in the introduction even if a bit out of scope. See for example quantile RF (DOI: 10.7717/peerj.5518 and https://doi.org/10.1016/j.envpol.2023.122501) and the area of applicability (https://doi.org/10.1111/2041-210X.13650).**

We mention NEX-GDM, which is a machine learning approach. We review the references and consider mentioning them for background.

Line 51: Spatial machine learning methods including Quantile Random Forest (QRF) can also be used to produce prediction intervals for uncertainty quantification (Hengl et al., 2018; Milà et al., 2023). However, this approach does not take into account spatial correlation and is sometimes combined with other geostatistical methods that do (Milà et al., 2023).

**The software (Julia?) and version used to perform the analysis should be included in the text.**

We will add a sentence to the methods explaining that all software version information (language and package) can be found in the Manifest.toml file provided on Github and in the archive on Zenodo.

Line 116: All calculations for this study were performed using the Julia programming language (Bezanson et al., 2017), and we make significant use of the GeoStats.jl package (Hoffimann, 2018) in particular. All package names and versions that were used can be found in the Manifest.toml file provided (see Code Availability).

**Referee 3**

We thank the referee for their time and attention. The referee raised several issues that were likely to be noted by other future readers of this paper. Addressing the referee's comments as we revise the manuscript will improve its clarity and, we believe, will make it appeal to a wider audience.

There were a few topics that came up multiple times in the review that we will address briefly here.

**1. Choice of methods for simulating Gaussian random fields:** We model the Tmax data as a stationary Gaussian random field. The Gaussian distribution has convenient properties that make it possible to solve explicitly for the mean and covariance matrix of a given conditional distribution, and then draw samples directly from that distribution. The LU method (Alabert, 1987) is one such way of doing this. The reason the LU method is often not used in practice is not because it is incorrect (it is sampling directly from the distribution of interest) but because it becomes computationally intractable as the scale of the problem grows. Other simulation methods like Turning Bands and Sequential Gaussian Simulation are computationally feasible methods of generating samples from high dimensional distributions that approximately reproduce the spatial covariance structure. Once we accept the model of a stationary Gaussian random field, then there is no theoretical (only practical) justification for using a method other than LU. From Goovaerts (1997; cited in the paper), "LU simulation should be the preferred Gaussian-based algorithm when many small realizations (few nodes) sparsely conditioned are to be generated." In this work, we generate many realizations of a small spatial area.

**2. Trend modeling:** The raw Tmax data are not stationary, but the model we have chosen assumes stationarity, so we need to apply a transformation. Trend modeling is a common practice in geostatistics that effectively models the data as being the sum of a deterministic component (the trend) and a random component (the random field). As we state in the paper, the trend is not something that can be observed objectively. From Pyrcz and Deutsch (2014; cited in the paper): "Any discussion on trends is subjective. All geostatisticians struggle with the separation of deterministic features, like trends, and stochastic correlated features, which are left to simulation." We have elected to use a very simple trend model that assumes Tmax varies (deterministically) linearly with elevation, latitude, and longitude. The trend model uses a linear regression that is fit locally for each station and prediction location. There is still spatial correlation in the detrended data. Per the reviewer's suggestion, we will add a figure that shows an example semivariogram, which will also demonstrate the spatial correlation present in the detrended data.

**3. The "correctness" of the predictive distributions:** We state three times throughout the paper (lines 67-68, 161-162, and 330-331) that there is no objectively correct predictive distribution and that uncertainty is a property of data and a model. After removing the trend and taking the normal score, the predictive distribution is theoretically correct with respect to the model (stationary Gaussian random field). The fact that the predictive distribution is

"theoretically correct" does not mean that it is actually correct in practice. This is why we do validation. The purpose is not to validate the theory of Gaussian processes. Rather, we are effectively testing whether it is reasonable to treat the data as stationary and Gaussian after performing various transformations.

**Specific comments**

**As far as I understand, the conclusion is based on two assumptions that I think limits the general conclusions in the paper: The main simplifying assumption is that realizations of Tmax is independent from one day to the next.**

This is true and we are also working on methods for quantifying uncertainty in meteorological time series (see https://meetingorganizer.copernicus.org/EGU24/EGU24-13593.html). However, we believe that the methods and analysis in the present work are significant enough to warrant publication. We will clarify this in the manuscript.

Line 205: At present, we do not model or try to exploit temporal correlation in Tmax or prediction errors. This topic will be addressed in future work.

**The second assumption is that the spatiotemporal trend in Tmax is sufficiently well modelled by regression. These two assumptions need to be addressed in the manuscript before publishing.**

See point 2 above. The trend estimation is performed independently for each day, so temporal trends do not need to be otherwise accounted for.

**I agree that kriging is a computationally efficient way to estimate the local mean, but the drawback is the underlying assumptions which require estimation of the trend and the Gaussian transformation (normal score transform). When it comes to assessment of estimation uncertainty and stochastic simulation, other techniques outperform the LU decomposition method. The most computational efficient method in geostatistics used to be the turning band method, TBM (Materhon, 1973). Application of TBM for spatiotemporal stochastic fields can be studied in Leblois and Creutin (2013).**

See point 1 above. Also, Turning Bands is a method that generates unconditional realizations. It requires an additional step to make the realizations conditional on the data (e.g., "conditioning by kriging"). We use a method that draws samples directly from the conditional distribution.

**In later years, Bayesian methods is developed to handle more general stochastic fields, and I would encourage the authors to study a method called "integrated nested laplace approximations" INLA (Rue et al., 2009). Application of INLA on non-stationary stochastic fields can be seen in Ingebrigtsen et al. (2014).**

Per the recommendation of referees 2 and 3, we will add references to Bayesian methods in the Introduction. Referee 2 also cited a paper using Gaussian Markov Random Fields (GMRF) and INLA which, unfortunately, does not include validation with respect to the accuracy of uncertainty quantification. The paper you cite does more to validate the probabilistic output of the model via CRPS. However, they apply the method to precipitation (not Tmax) so we cannot do a direct comparison. Comparing the results using our method with results using Bayesian approaches would be interesting, but would be a significant undertaking and is beyond the scope of this work.

Line 54: Finally, Bayesian methods using Gaussian Markov Random Fields (GMRF) have also been used to perform probabilistic interpolation of meteorological data. For example, Fioravanti et al. (2023) applied these methods to air temperature (but did not quantitatively validate the uncertainty quantification) and Ingebrigtsen et al. (2014) applied them to precipitation data. In future work, it would be instructive to compare predictive distributions produced using QRF and Bayesian GMRF-based methods with those produced using the more classical geostatistical methods described in this work.

**Time series of Tmax for 2022 for at least one station. It would be interesting to see a timeseries representing typical coastal areas, inland areas and maybe mountain areas. This may be included as part of Fig.1 without requiring too much extra space.**

We can add example time series to Fig. 1, as suggested.

See the updated Fig. 1.

**Method and results of the trend analysis should be included. What kind of regression is used to estimate the trend? What kind of assumptions are made? What kind of data is included? I understand the Digital Elevation Map (DEM) is used, but what is the cross-correlation between the Tmax and the DEM? Is there a seasonality in the trend, or is it sufficient with an average (steady-state) trend?**

See point 2 above. The purpose of the trend model is to account for correlation between Tmax and elevation (as well as lat and lon). We will add text to the methods clarifying that the trend, normal score transformation, and semivariogram are calculated for each day.

Line 196: Model fitting is performed using the detrended and normal scored data (the data in Figure 2E-F) for each day independently (i.e., there is a different semi-variogram model for each day).

**Why is not the shortest distance to the coast included in the trend analysis?**

We tried various approaches for this, but it did not improve results using our linear trend model. Topography is also important and it was difficult to account for this effect without significantly increasing the complexity of the trend model. For example, a station in a valley very near the

coast may be much warmer than a station that is further away but has no hills/mountains in between. More generally, the trend modeling is not meant to exhaustively account for all sources of variation. We will add this explanation in the revised paper.

Line 142: The purpose of detrending is not to explain all variation in Tmax values. Rather, the purpose is to control for large scale variation such that the residuals, after detrending, can be modeled as a random field whose mean does not vary deterministically in space. While it could be useful to incorporate other covariates (e.g., distance to the ocean), doing so would likely require a nonlinear trend model.

**A figure of the experimental and the theoretical semi-variogram function (Eq.1). I would suggest making a figure based on raw Tmax data (engineering values), and a figure based on transformed data (detrended and normal score transformed data). A table which gives the estimated values for the parameters in Eq.1 (i.e. s, n, and r) for raw data and transformed data would add value to the article.**

We will add a figure showing an example semivariogram (also recommended by Referee 2). We estimate theoretical semivariograms for each day, so we will pick some representative days from different seasons.

See the new Supplementary Information.

**An intuitive way to visualize the cross-validation results is to calculate the probability score of the left-out observation. For each left-out observation, a conditional probability density function is derived as the authors clearly show. The probability score is the mapping of the left-out observation on the estimated conditional cumulative probability function (similar to Fig.3, but where the density f(X), is integrated from min to max and normalized to 1). The probability score can be shown as a histogram for all probability scores for one day, or for all days lumped together. A 'perfect' cross-validation result would have a very narrow distribution centered around a probability score equal to 0.5. In case the distribution is centered away from 0.5, the estimates are biased. If the distribution is wide, the estimates are not very precise.**

 the main purpose of the validation exercise presented here is to test how well the theoretical coverage of the predictive distributions match the actual coverage in practice. The suggested validation method would not capture this. Probability scores clustered around 0.5 would indicate the predictive distributions are under-confident.

**I suggest that histograms of probability scores is done on Tmax observations (engineering values) and not on the transformed data (in the computational domain).**

The validation statistics are calculated using the back-transformed values. As described in section 2.3, the procedure includes reverting the normal score transformation and adding in the estimated trend.

**I think histograms of probability scores would complement the accuracy criteria given in Eq.2 and fig. 4.**

Rank/score histograms would convey approximately the same information as the scatter plots in Fig. 5. Using the prediction interval validation approach allows us to compute summary statistics for each day, as shown in Fig. 4.

**Abstract needs to be revised**

We will revise the abstract (per suggestion of all three referees). Done.

**What exactly is meant by "Conventional point estimate error statistics are not well-suited to describing spatial and temporal variation in the accuracy of spatially interpolated meteorological variables."? (line 8-9). The study is based on conventional statistics for point estimation. The phrase indicates that grid averages might be considered, but grid averages (block-kriging) versus point estimation is not a part of the work. The conclusion (line 24-26) should be more precise. The phrase: "While kriging and related spatial regression methods have previously been used for meteorological data interpolation, they have only been used to produce gridded point estimates." (Line 62-64) needs to be rephrased. It's not clear to me what the authors indicate. Given the assumption in kriging, geostatistics provide a conditional probability density function in all points of estimation.**

In this context, a "point estimate" refers to an estimate that is a single value, as opposed to a prediction range or distribution. We will clarify this distinction in the abstract and body of the manuscript.

We have clarified what is meant by a point estimate in the abstract and introduction.

**The next phrase ("…enable us to compute theoretically correct predictive distributions at each prediction location." Line 64-65), indicate that the authors have developed new theory, but that is not true.**

See point 3 above. We explicitly state in the abstract and introduction that we are applying established methods and do not claim to have developed a new theory.

We deleted this sentence to avoid confusion.

**The authors does not give any theory on how to estimate a 'correct' trend, and the normal score transform is not a 'theoretically correct' transformation. It is a pragmatic**

**way to achieve Gaussian distribution, and it is limited by the empirical data (as the authors correctly indicate later!).**

See points 2 and 3 above. We do not claim that the trend model or the normal score transformation are "correct."

**These two points (the assumptions of stationarity and Gaussian distributions) is addressed in Bayesian geostatistics, which I encourage the authors to acknowledge!**

We will add references to Bayesian methods in the introduction.

See comment above with inclusion of references to Bayesian methods.

**Section 2.2 needs revision. The method for trend analysis is not presented in detail. What is "… regressing …" Tmax and elevation? If linear regression is done in a local neighborhood, the residual is assumed to be independent. In that case there will be no spatial correlation in the residual, and the kriging variance can not capitalize on the correlation structure. If there is a correlation structure in the residual, which the authors indicate, there is a logical inconsistency of doing a linear regression. It is not uncommon practice to estimate a trend by linear regression, but in that case, is should be clearly stated, and not expressed as a "… theoretically correct …" procedure.**

See point 2 above. To "regress y on x" means to estimate parameters a and b for the linear model y = ax + b. We do not describe the full procedure as theoretically correct and repeatedly acknowledge that there is no objectively correct uncertainty quantification.

The revised methods section now includes a regression specification.

**I would also like to see a motivation for using the conditional stochastic simulation method (CGS) suggested by Alabert (1987). A number of alternative simulation algorithms are available, and I think the readers need a good reason for neglecting more recent advances.**

See point 1 above.

Line 175: The LU method samples from the exact multivariate Gaussian predictive distribution. For larger scale simulations, the LU method becomes computationally infeasible and other algorithms and approximations may be required. However, in this example, we are drawing samples for a small number of locations so the LU method works well.

**The phrase "… CGS produces realizations that are spatially coherent with respect to the model of spatial covariance …, whereas gridded estimates produced by OK do not." (Line 103-104) needs clarification. Please, explain clearly the difference in the results of**

**CGS and OK with respect to simulation and estimation of the (unknown) conditional probability density function!**

We will revise the methods section to more explicitly that CGS can be used to describe the joint spatial predictive distribution and kriging only expresses marginal distributions. This is illustrated in Fig. 7 and addressed in the Discussion.

The revised Methods section should make this clearer.

**Fig. 2 illustrates the procedure in a good way and should be included, but I think it belongs to the results section.**

The purpose of Fig. 2 is to illustrate the methods. Nothing in the figure is presented as a result.

**I would also suggest using the same coordinate system in Fig. 1 and 2., preferably (Long, Lat) instead of (km Easting, km Northing).**

We will standardize the units on figures.

All maps are now displayed in lat/lon coordinates.

**The figure text in Fig. 6 needs revision. The term "prediction interval" indicates "p +/- 1", which is misleading. In line 225-228, it is clear that it's the percentile between 0.75 and 0.25 that is meant for the 0.5 percentile and 0.95 and 0.05 for the 0.9 percentile.**

The definition of prediction interval is given in section  2.3. The value of the interval corresponds to its theoretical coverage level.

**The comments regarding the x- and y-label as for Fig. 1 and Fig. 2 is also valid for Fig. 6, se (Long, Lat) for all figures!**

See above.

**Looking carefully at Fig. 6, I recognize an anomaly in the median results (left panel in Fig. 6) close to Long = -122, Lat = 42 (in Northern California). I recommend the authors to examine the observations in that area, maybe there is a station in that area which deviate from the nearest neighboring stations?**

On closer inspection, I believe what the reviewer is referring to is Mount Shasta. The pattern in the Tmax maps closely matches that of the DEM. There is a significant local minimum in Tmax due to the local maximum in elevation. An annotated DEM is shown here.

[Figure]

However, it is also the case that the weather station closest to Mount Shasta has an issue. The elevation is listed as 0 m in the GHCN metadata, which is clearly not correct. The measured Tmax values are not so different from surrounding stations, but the trend calculation was artificially depressing the value (assuming the Tmax was observed at sea level). We found one other station with this issue. (We reported the issue to NCEI.) For these stations, we instead use the elevation from the DEM in all calculations.

**I find the manuscript interesting to read and well written, but I encourage the authors to elaborate the literature review and include some more results (as indicated above) before the manuscript is published. I think the conclusion needs to be revised. It is not clear to me what is meant by "We conclude that spatial correlation in Tmax errors is relatively small," (line 24). If it means that the estimation uncertainty cannot be improved by including more information, I do not agree. The conclusions should be deduced directly from the discussion. To me this is not evident in the current version. I think it is possible to draw more precise conclusions based on the material and the work presented in the manuscript.**

Regarding "cannot be improved by including more information": that is not a conclusion we make. In the introduction, we say that including additional information improves estimates in PRISM and NEX-GDM products. We will revise this section to make the conclusions clearer.

We modified the discussion of CGS to be more precise.

---

## Referee Report (RR1)

**Review comments on EGUsphere manuscript:**

https://doi.org/10.5194/egusphere-2024-1886, revised version

Title:

A Method for Quantifying Uncertainty in Spatially Interpolated Meteorological Data with Application to Daily Maximum Air Temperature

Authors: Conor T. Doherty, Weile Wang, Hirofumi Hashimoto, Ian G. Brosnan

The revised version of the manuscript cannot be recommended for publishing in the Geoscientific Model Development. The authors do not present any new methods or geostatistical theory, and the cross-validation results is not presented in a clear and transparent way. Below, is a summary of my main critical points:

1) The title is very misleading. It gives the impression that a new method is presented for "… quantifying uncertainty in spatially interpolated meteorological data …". The content of the paper is a case study of Tmax data from California (c.f. line 180-182), no "… comprehensive evaluation of spatial uncertainty quantification" is presented. Thus, the title of the manuscript is not consistent to the content.

2) The method is not mathematically clearly presented. In classical regression (1), the residual is assumed to be spatially uncorrelated, but the semivariogram analysis demonstrates that the assumption is not correct. This logical flaw is not discussed in the paper. The uncertainties in the regression parameters are not quantified in the analysis.

3) It's not clear to me whether a regression is applied for all days, or only for the yearly average of Tmax for each station. The same is true for the normal score transform. Is there a "lumped" normal score transform, or is the normal score done for each day? From figure 9 (in supplementary material) it looks like it's a lumped normal score, but a semivariogram is applied for every day.

4) The application of conditional Gaussian simulation (CGS) is not sufficiently explained. What parameters are used, how many realizations is generated?

5) Cross-validation is not explained in sufficient detail. If the authors decide to rework the manuscript, I recommend Continuous Ranked Probability Score (CRPS; Gneiting & Raftery 2004).

6) The authors underline that uncertainties are "overlooked" (line 8) and "… relatively little attention is paid to the errors …" (line 25). This is not true. Uncertainties are a core issue in geostatistical analysis also in meteorological studies (c.f. Lensen et al. 2019; Lussana et al. 2018).

7) Figures are missing (c.f. Fig.3 transparent lines are missing, Fig.4, Fig. 8 are not shown). I consider this as a minor technical mistake, but I've checked several times for an update, and I've also used different web-browsers to make sure it's not my mistake.

**References:**

Lenssen, N.J.L., Schmidt, G.A., Hansen, J.E., Menne, M.J., Persin, A., Ruedy, R. et al. (2019) Improvements in the gistemp uncertainty model. Journal of Geophysical Research: Atmospheres, 124(12), 6307-6326.

Lussana, C., O.E. Tveito, F. Uboldi (2018) Three-dimensional spatial interpolation of 2 m temperature over Norway. Quarterly Journal of the Royal Meteorological Society 144 (711), 344 364

Gneiting, T. & Raftery, A. E. (2004) Strictly Proper Scoring Rules, Prediction, and Estimation. Technical Report No. 463. Department of Statistics, University of Washington, Seattle, Washington. doi:10.1198/016214506000001437.